# An Optical Power Limiting and Ultrafast Photophysics Investigation of a Series of Multi-Branched Heavy Atom Substituted Fluorene Molecules

**Hampus Lundén [1,2], Delphine Pitrat [3], Jean-Christophe Mulatier [3], Cyrille Monnereau [3], Iulia Minda [4], Adrien Liotta [3], Pavel Chábera [4] , Didrik K. Hopen [5], Cesar Lopes [1], Stéphane Parola [3], Tönu Pullerits [4], Chantal Andraud [3] and Mikael Lindgren [2,5,*]**

[1] FOI, Swedish Defence Research Agency, Olaus Magnus väg 42, 583 30 Linköping, Sweden; hampus.lunden@foi.se (H.L.); cesar.lopes@foi.se (C.L.)
[2] IFM-Department of Physics, Chemistry and Biology, Linköping University, 581 83 Linköping, Sweden
[3] Laboratoire de Chimie, Univ Lyon, ENS de Lyon, CNRS UMR 5182, Université Claude Bernard Lyon 1, F69342, Cedex 07 Lyon, France; delphine.pitrat@ens-lyon.fr (D.P.); jean-christophe.mulatier@ens-lyon.fr (J.-C.M.); cyrille.monnereau@ens-lyon.fr (C.M.); liotta.adrien@gmail.com (A.L.); stephane.parola@ens-lyon.fr (S.P.); chantal.andraud@ens-lyon.fr (C.A.)
[4] The Division of Chemical Physics and NanoLund, Lund University, Box 124, 22100 Lund, Sweden; iulia.minda@chemphys.lu.se (I.M.); pavel.chabera@chemphys.lu.se (P.C.); tonu.pullerits@chemphys.lu.se (T.P.)
[5] Department of Physics, Norwegian University of Science and Technology, N-7491 Trondheim, Norway; didrikkh@stud.ntnu.no
* Correspondence: mikael.lindgren@ntnu.no; Tel.: +47-414-66-510

**Abstract:** A common molecular design paradigm for optical power limiting (OPL) applications is to introduce heavy atoms that promote intersystem crossing and triplet excited states. In order to investigate this effect, three multi-branched fluorene molecules were prepared where the central moiety was either an organic benzene unit, para-dibromobenzene, or a platinum(II)–alkynyl unit. All three molecules showed good nanosecond OPL performance in solution. However, only the dibromobenzene and Pt–alkynyl compounds showed strong microsecond triplet excited state absorption (ESA). To investigate the photophysical cause of the OPL, especially for the fully organic molecule, photokinetic measurements including ultrafast pump–probe spectroscopy were performed. At nanosecond timescales, the ESA of the organic molecule was larger than the two with intersystem crossing (ISC) promoters, explaining its good OPL performance. This points to a design strategy where the singlet-state ESA is balanced with the ISC rate to increase OPL performance at the beginning of a nanosecond pulse.

**Keywords:** optical power limiting; excited state absorption; reverse saturable absorption; ultra-fast pump–probe spectroscopy

## 1. Introduction

Organic and organometallic dyes with nonlinear optical properties are successfully being used for self-activated optical power limiting (OPL) [1–3]. A good OPL material shows low linear attenuation, but at higher intensities/fluences above a certain "cut-off level", the optical radiation is efficiently attenuated [1–3]. The search for new dyes with increased nonlinear performance and linear transmittance is ongoing in different ranges of wavelengths [1–8]. To maximize the OPL, several

optically nonlinear processes are combined: Two-photon absorption (2PA) in conjunction with Excited State Absorption (ESA), i.e., three-photon absorption [1–3]. For that purpose, a heavy-metal atom, such as platinum, is often used to facilitate efficient intersystem crossing (ISC) into a highly absorbing and long-lived triplet state [1–3]. The heavy-metal atom(s) are often placed in the center of and/or at the edges of large molecular complexes with π-conjugated 2PA chromophores [2]. Examples of heavy-metal atoms used in organometallic dyes include platinum, gold, mercury, ruthenium, and palladium [1–10]. Among pure organic ISC promoters para-dibromophenyl has also been used for singlet $O_2$ generation in applications of photodynamic therapy [11,12].

A popular moiety in π-conjugated 2PA chromophores is fluorene, selected for its high 2PA [1,9,13–17] in the visible wavelength range. Apart from the favorable 2PA, singlet excited state absorption (ESA) has also been reported for fluorenes [1,15,18], e.g., 2,2′-(9,9-dihexyl) bifluorene showed a relatively long-lived singlet excited state of 450 ps and a 2PA peak of around 540 nm [15]. The two alkyl chains on the fluorene unit are included for increased solubility [19]. This is especially important if the dyes are to be incorporated into a solid matrix for practical applications [20–23]. Multibranching geometries have been used to increase the 2PA of the fluorene molecule [7,19,24]. The advantages of a multibranched geometry go beyond the ability to include more 2PA fluorene units, as it also opens up the possibility to pursue additional design goals [19].

The three molecules investigated in this work are shown in Figure 1. They are all based around the same multi-branched molecular geometry. Each molecule includes (2 × 3) 9,9-dihexylfluorene moieties on the 1,3 and 5 position of an o-benzene ring. The central moiety, connected to two of the fluorene moieties in a para-configuration, is either a benzene ring (A), a para-dibromobenzene unit (B), or a platinum(II)-alkynyl unit (C). Furthermore, efficient 2PA chromophores have traditionally been built around a π-conjugated moiety separating an electron donor and/or acceptor [1–3]. In platinum(II)–alkynyls, the platinum center has been known to act as an electron donor [25], while the dibromobenzene ring can be expected to behave as an acceptor due to the electronegativity of bromine [26]. Platinum was selected for its strong spin–orbit coupling, allowing for efficient ISC [2,17,27]. The four-coordinate platinum(II) center is often combined with two phosphine ligands and two π-conjugated aryl acetylide ligands [2], as in one of the molecules of this work.

a) Pd(PPh$_3$)$_4$, toluene/Na$_2$CO$_3$ 1N; b) TMSA, PdCl$_2$(PPh$_3$)$_2$, CuI, THF/Et$_3$N; c) K$_2$CO$_3$, MeOH/THF; d) PtCl$_2$(PBu$_3$)$_2$, CuI, Et$_3$N

**Figure 1.** Molecule "A", "B", and "C". (**a**) Pd(PPh$_3$)$_4$, toluene/Na$_2$CO$_3$ 1N; (**b**) TMSA, PdCl$_2$(PPh$_3$)$_2$, CuI, THF/Et$_3$N; (**c**) K$_2$CO$_3$, MeOH/THF; (**d**) PtCl$_2$(PBu$_3$)$_2$, CuI, Et$_3$N.

The purpose of this study was to investigate the effect that different ISC promoter moieties have on the OPL properties of chromophores. Replacing expensive (and sometimes toxic) metals such as platinum would be advantageous. Molecule "A" was selected as a fully organic baseline without an ISC promoter. The para-dibromobenzene group in molecule "B" and the platinum(II)–alkynyl unit in molecule "C" plays the role of ISC-promoting moiety. The para-dibromobenzene group in molecule "B" was selected for the heavy-atom effect related to bromine substituents, which was shown to lead to efficient ISC when appropriately localized in the molecule [11,12,26]. The relatively similar nanosecond OPL performance of the three molecules required a deeper photokinetic investigation with pump–probe ultrafast spectroscopy. The transient absorption spectra and excited state lifetimes were found to give a qualitative explanation of the results.

## 2. Results

For simplicity, we abbreviate the molecules "A", "B", and "C", where "A" represents the all-organic molecule, "B" the molecule with the dibromine central unit, and finally, "C" the molecule with the platinum acetylide central unit.

### 2.1. Luminescence and Linear Transmittance

The molar attenuation spectra of the three molecules are presented in Figure 2 (left panel). The blue/UV absorption edge wavelength is the longest for "C" and shortest for "B".

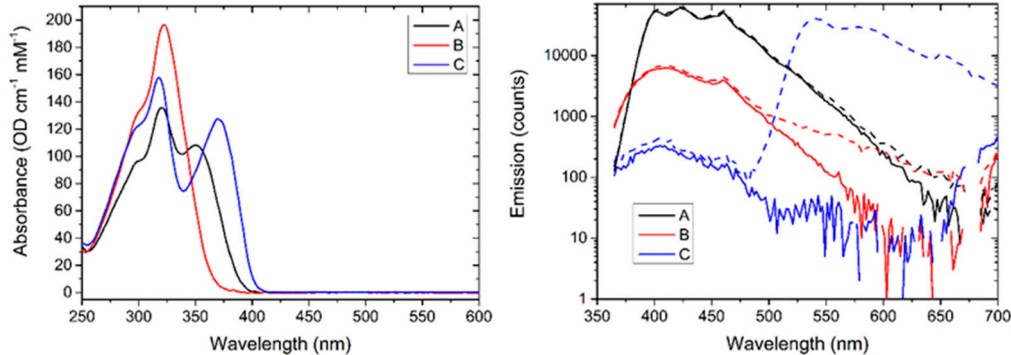

**Figure 2.** (**Left Panel**) Absorbance spectra of molecules "A" (black), "B" (red), and "C" (blue). (**Right Panel**) Emission spectra of molecules "A" (black), "B" (red), and "C" (blue). The dashed lines are after oxygen removal through argon bubbling. The excitation wavelength was 337 nm.

The luminescence spectra for deoxygenated (Ar purged; dashed curves) and pristine samples (solid lines) using THF as solvent are shown in Figure 2 (right panel). Note that the scale is logarithmic. Upon excitation at 337 nm, molecule "A" showed an emission several orders of magnitude larger in the 400–475 nm wavelength region than both "B" and "C", originating from the fluorene units. After purging with Ar-gas molecules, "C" showed, as is well known for Pt–acetylide complexes, strong phosphorescence in the 500–650 nm region. It should be pointed out that the fluorescence from the "B" molecule is also very weak in comparison to that of "A". Some extremely weak phosphorescence could be detected for the "B" molecule. Thus, "C" showed the largest changes due to oxygen quenching, followed by "B". Molecule "A" showed no oxygen quenching except for a very weak tail near 650 nm.

The fluorescence lifetimes for "A" and "B" diluted in THF solution were found to be 550 ps and 180 ps, respectively, using the TC-SPC (time-correlated single photon counting) measurement technique [28]. The phosphorescence lifetime of "C" of THF solvent purged with Ar-gas for 8 min was found to be 340 μs. The sensitivity of phosphorescence lifetimes to oxygen concentration for a series of Pt–acetylide compounds was discussed previously [29].

The 2PA cross-section of "A" was measured by comparing the integrated TPA-induced fluorescence vs. fluorescein (pH 11), using single-photon counting at 1 MHz [27–29]. The samples were examined with the same concentration of 25 μM. The 2PA for molecule "A" remained fairly constant in the NIR region with the small cross-section never exceeding 4 GM in the range 725–800 nm. At longer wavelengths, it was too weak to be detected with the available setup. The TPA cross-section for the phosphorescent molecules "B" and "C" could not be examined using our laser system as even the low pulse repetition frequencies could yield unwanted excited state absorption of the relatively long-lived triplet states.

### 2.2. Wavelength-Scanned OPL Measurements

To compile the results of the wavelength-scanned OPL measurements, the cut-off level was estimated at each wavelength. This was done by doing a linear fit of the data points above an input fluence of 20 J/cm$^2$ and reporting the estimated transmitted pulse energy at 25 J/cm$^2$. The results are

shown in Figure 3. The platinum(II)–alkynyl molecule, "C", showed the best performance, followed by the purely organic molecule "A". All three molecules showed similar performance around 550 nm. In the 450 nm to 500 nm wavelength range, molecule "C" showed a better cutoff level than the fairly similarly performing molecules "A" and "B". At wavelengths above 600 nm, the cutoff level of the molecules showed a marked difference. The dibromophenyl-based molecule "B" had the highest cutoff level, while molecule "C" showed the best performance.

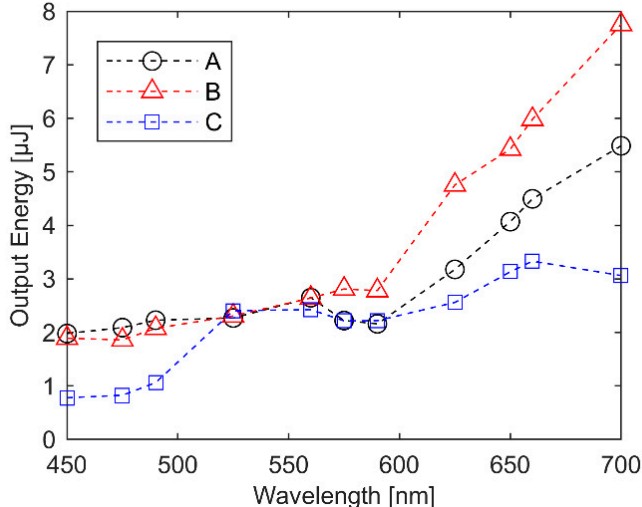

**Figure 3.** Optical power limiting (OPL) spectra of molecules "A" (black circles), "B" (red triangles), and "C" (blue squares). The output energy was calculated by doing a linear fit of every point above 20 J/cm$^2$ input fluence and taking the resulting value at 25 J/cm$^2$ input fluence.

OPL measurement plots for each wavelength, including their linear fits, are shown in Figures S1–S6. For some wavelengths, especially for molecule "B", the output energy showed a dip at the lower fluence levels of the nonlinear regime (see Figure 4). Measurements on cuvettes with only CH$_2$CL$_2$ solvent showed no nonlinear effects.

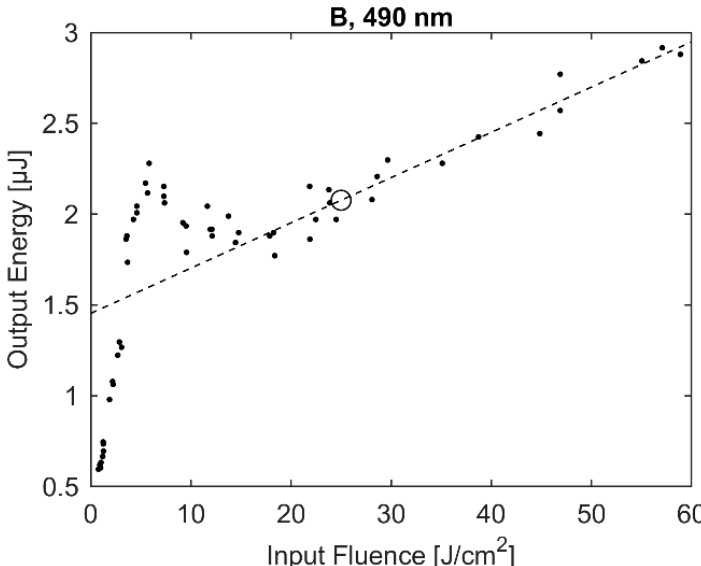

**Figure 4.** Representative OPL measurement for molecule "B" at 490 nm excitation wavelength. The dots are the measured data. The dashed line is the linear fit of the "cutoff level", and the circle is the value used for plotting, as in Figure 3.

### 2.3. Excited State Absorption

The microsecond ESA measurements are shown in Figure 5. The plots are for the largest ESA recorded with zero delay between the ns pump and the white probe flash. In the calculation of the ESA spectrum, the optical absorption spectrum of the unpumped sample was used, giving the negative UV absorption features of samples "B" and "C" in particular, which had long-lived triplet states [30]. For *microsecond* ESA, molecule "B" showed the highest absorption of the triplet state in the 600–700 nm region, followed by molecule "C". Both the ground state bleaching and ESA were relatively weak for molecule "A", compared to that for "B" and "C", meaning that most of the excited molecules were relaxed to the ground singlet state within the μs excitation pulse.

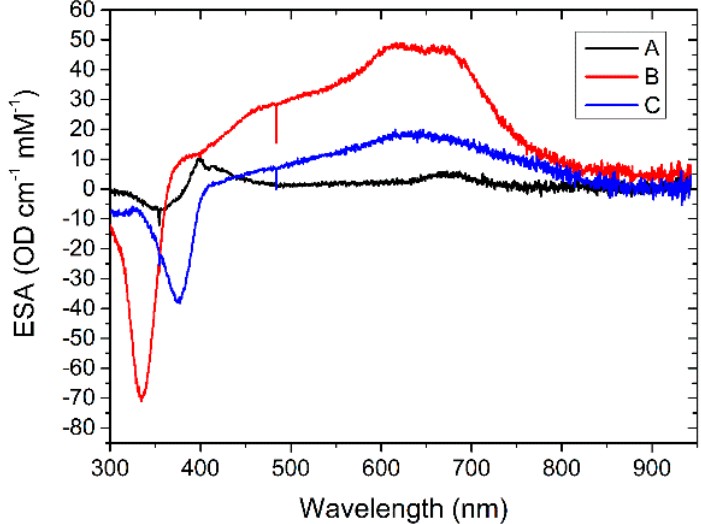

**Figure 5.** Excited State Absorption (ESA) spectra of molecules "A" (black), "B" (red) and "C" (blue).

Ultra-Fast Transient Absorption Measurements

The ultra-fast transient absorption measurements and their curve-fits are shown in Figures S7–S9. Curve-fitting was done by an iterative use of a simplex method [31] for the decay times and linear least squares for the amplitudes. The model used was the standard decay/grow-in model [32].

$$OD(\lambda, t) = \sum_i c_i(\lambda) e^{-t/\tau_i} + d(\lambda) \tag{1}$$

The spectral basis vectors, $c_i(\lambda)$ for decay time $\tau_i$, are presented in Figures 6–8. $d(\lambda)$ represent both decays too slow to resolve and the constant part of grow-ins.

Spectral basis vectors of molecule "A" are shown in Figure 6. Three decay times were identified as 2.7 ps, 37.2 ps, and 545 ps. The long-lived 545 ps lifetime was an order of magnitude stronger in absorbance than the other decays and showed a negative signal towards the blue wavelengths.

The spectral basis vectors for molecule "B" are shown in Figure 7. Decays were measured to occur with time constants of 7.2 ps and 87.6 ps. Moreover, an ultrafast component with a time constant in the order of our temporal resolution, as well as a much slower >1 ns component, was also obtained. While these decays are observed in the experimental results, the lack of data points at these time ranges induce large uncertainties in the fits. As a result, the *d* term was removed to avoid degeneracy issues in relation to the slow nanosecond decay. Finally, the 87.6 ps decay is characterized by a negative component towards the blue wavelengths, similar to the 545 ps decay observed for molecule "A".

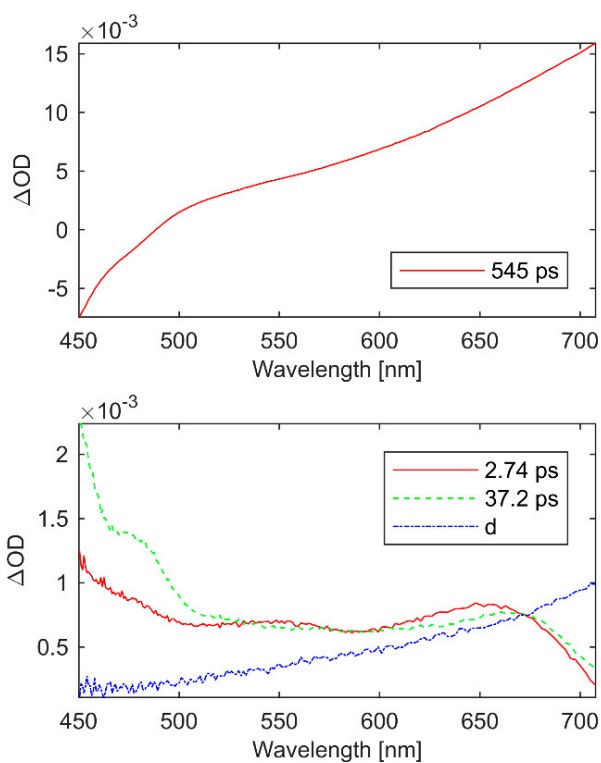

**Figure 6.** Spectral basis vectors for molecule "A". See Equation (1) for explanation.

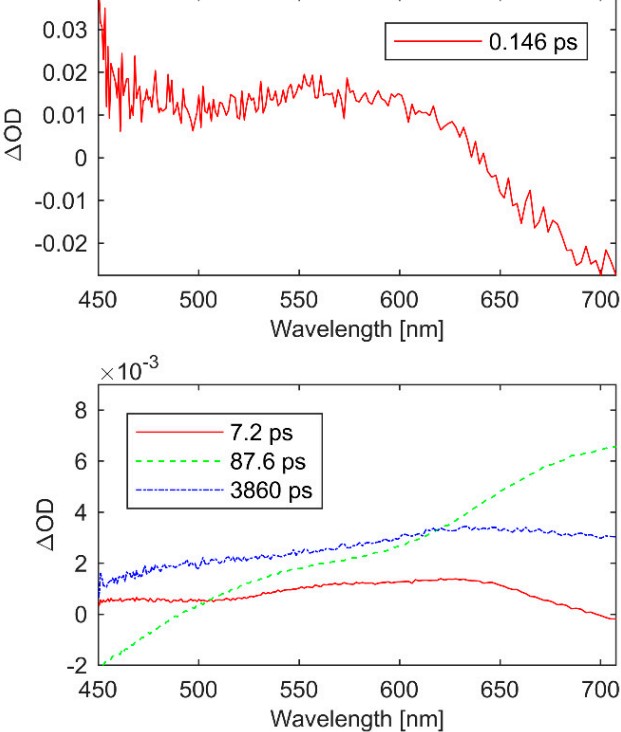

**Figure 7.** Spectral basis vectors for molecule "B". See Equation (1) for explanation. While the optimizer converged for the 0.15 ps and 3.9 ns decay, too few data points were available to give a trustworthy fit.

For molecule "C", the basis vectors are presented in Figure 8. Decays with time constants of 0.2 ps, 2.6 ps, 70.7 ps, and 710 ps were obtained. The quality of the subpicosecond decay fit was, as with molecule "B", of limited quality. Furthermore, while the 2.6 ps decay was characterized by a negative

amplitude near 450 nm, it was not as prominent as for the 545 ps and 87.6 ps decays obtained for molecules "A" and "B".

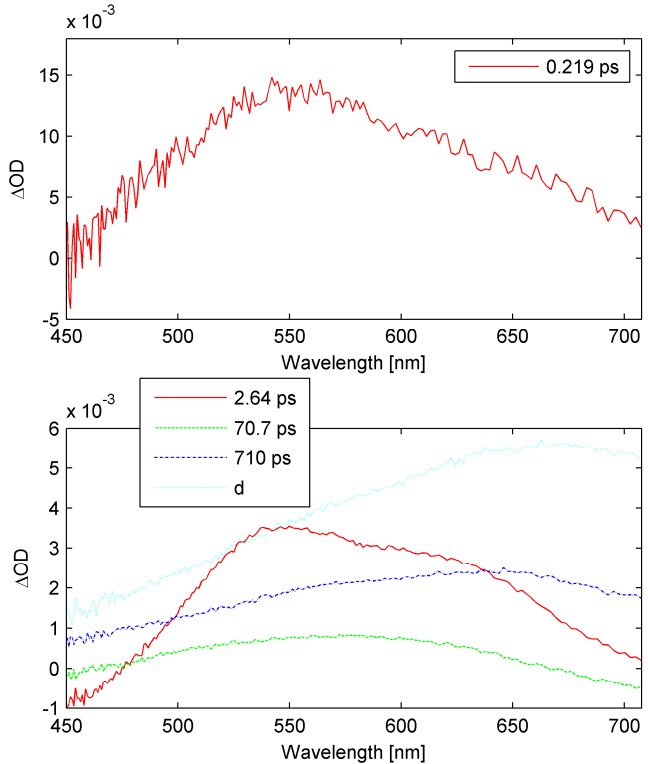

**Figure 8.** Spectral basis vectors for molecule "C". See Equation (1) for explanation. While the optimizer converged for the 0.22 ps decay, too few data points were available to give a trustworthy fit.

## 3. Discussion

The optical power-limiting measurements in Figure 3 show that all three molecules showed good OPL performance through the visible wavelength range. The better performance of molecule "C" towards the blue wavelength range was expected due to the red-shifted linear absorption edge (Figure 2). The lowered cutoff level towards 700 nm for molecule "C" is probably explained by 2PA. Platinum(II)–alkynyl complexes are known to derive their OPL performance in the red wavelength region from 2PA population of the excited states [25,27,33].

Considering the temporal laser pulse length (3–5 ns), the relatively long lived triplet state was expected to play an important role in OPL performance, as in similar Pt–alkynyls [34]. It has previously been reported for Pt–alkynyls that its triplet state can be quenched by oxygen [20]. In the emission spectra shown in Figure 2 (right panel), molecules "B" and "C" demonstrated a sensitivity to singlet oxygen quenching, but the impact of oxygen removal on molecule "A" solution was low. Secondly, the experiments showed that both the ground state bleaching and ESA of molecule "A" were low for microsecond ESA (Figure 5). This indicates that the OPL performance of the fully organic molecule "A" does not depend on triplet state ESA. Morel et al. [15] found that a small fluorene molecule, 2,2'-(9,9-dihexyl) bifluorene, showed both a pronounced 2PA efficiency in the visible wavelength region and a singlet exited state of 450 ps. The ultrafast spectroscopy measurements in Figure 6 show a relatively long-lived decay of 545 ps. Around 450 nm, this decay is characterized by a negative ΔOD, which we assign to luminescence. Together with the measured fluorescence decay of 550 ps (in THF), this indicates that this level is the S1 state. This state is sufficiently long-lived to play a relevant role in OPL performance for nanosecond pulses. Furthermore, its ΔOD is large, compared to the ESA of the other molecules.

The 87.6 ps decay component characteristic to molecule "B" shows similarities in spectral contribution to the 545 ps decay of molecule "A", including the negative ΔOD around 450 nm. This similarity and that the decay is in the same lifetime range as the measured fluorescence lifetime for molecule "B" in THF of 180 ps make it possible to assign this decay to the S1 state. We interpret this as the Br atoms facilitating intersystem crossing, while shortening the lifetime of the S1 state. This finding is in agreement with earlier works where the dibromophenyl moiety was used as an ISC promoter for singlet oxygen generation by 2PA chromophores [11,12]. All three molecules show a complex decay structure, especially molecules "B" and "C". Further ultrafast spectroscopy studies, e.g., ground state bleaching, alternative excitation wavelengths and larger timespans, are necessary to build meaningful and complete photokinetic models for these molecules. The decay complexity of molecules "B" and "C" indicate the presence of singlet and/or triplet manifolds.

Fluorene molecules (linear oligomers and branched molecules) have shown interesting 2PA in the visible wavelength range [15,24]. For the multibranched fluorene molecules reported in this paper, a large number of fluorene units were assembled to raise the total 2PA cross-section. Overlapping ESA and 2PA spectra have been shown to result in increased OPL performance through three-photon absorption (2PA + ESA) [35]. The low correlation between the microsecond ESA spectra (Figure 6) and OPL spectra (Figure 4) is indicative of 2PA being important for explaining the wavelength dependence of the OPL response. Femtosecond Z-scan measurements in the visible wavelength region could elucidate the impact of 2PA vs. ESA on OPL performance for these molecules.

The dip in the OPL measurements at intermediate fluences around 20 J/cm$^2$, as shown in Figure 4, point to a nonlinearity in populating the excited state. While molecule "A" showed some tendency of this pattern, it was most pronounced for molecule "B". While further studies are necessary to find the cause of this dip in transmitted pulse energy, sample damage is improbable due to the molecules being in the solution.

Conclusively, all three molecules showed good OPL performance through the visible spectrum, making all of them candidates for both incorporation into glass matrixes and for further studies. While direct comparisons across different measurement series with the *f*/5 setup are problematic [36], the cutoff levels of these molecules compare favorably to PE2, e.g., [37].

The results of this study also highlight the importance of the interplay between singlet ESA and triplet ESA, even for nanosecond pulses. Molecule "A" showed better OPL performance than molecule B. While the impact of 2PA is unknown, the results hint that the depopulation of the singlet state into the triplet state sometimes can be counterproductive, even for nanosecond pulses. This is especially relevant considering that it is the beginning of the pulse that is of most importance for OPL performance [36,38].

## 4. Materials and Methods

THF solutions of approximately 1–5 µM were prepared for luminescence spectra, luminescence lifetimes, and linear absorbance, and microsecond excited state absorption measurements. The OPL measurements were performed using 25 mM CH$_2$Cl$_2$ solutions in 1 mm quartz cuvettes. The femtosecond transient absorption measurements were done on 25 µM CH$_2$Cl$_2$ solutions in 2 mm cuvettes.

### 4.1. Synthesis

The full synthesis and characterization of intermediate species and ligands for complex "C" are reported elsewhere [39] as well as the detailed synthesis and characterization of "C" [40].

### 4.1.1. Synthesis of "A"

To a degassed solution of 7-((3,5-bis(9,9-di-*n*-hexylfluoren-2-yl)benzen-1-yl)-9,9-di-*n*-hexylfluoren-2-yl)boronic acid (100 mg, 0.0893 mmol) and 1,4-diiodobenzene (13.4 mg, 0.0406 mmol) in toluene (3 mL), a degassed solution of Na$_2$CO$_3$ 1 M in water (3 mL) and tetrakis(triphenylphosphine) palladium(0)

(1.8 mg, 0.0016 mmol) was added. The mixture was stirred at 90 °C for 48 h. After cooling to room temperature, the layers were separated, and the aqueous layer was diluted and extracted with $3 \times 20$ mL of dichloromethane. The combined organic layers were washed with brine, dried over $Na_2SO_4$, filtered, and concentrated under reduced pressure to produce 104 mg of brown oil. The crude was purified by flash chromatography on silica gel (9/1 petroleum ether/$CH_2Cl_2$) to provide 67 mg (74%) of white solid. $^1$H NMR (300 MHz, CDCl$_3$): 7.93 (s, 6 H), 7.89–7.66 (m, 32 H), 7.39–7.33 (m, 12 H), 2.08–2.02 (m, 24 H), 1.10–1.08 (m, 72 H), 0.79–0.75 (m, 60 H).

### 4.1.2. Synthesis of "B"

To a degassed solution of 7-((3,5-bis(9,9-di-*n*-hexylfluoren-2-yl)benzen-1-yl)-9,9-di-*n*-hexylfluoren-2-yl)boronic acid (150 mg, 0.134 mmol) and 1,4-dibromo-2,5-dibromobenzene (26 mg, 0.054 mmol) in toluene (3 mL), a degassed solution of $Na_2CO_3$ 1 M in water (3 mL) and tetrakis(triphenylphosphine) palladium (0) (2.4 mg, 0.002 mmol) was added. The mixture was stirred at 90 °C for 48 h. After cooling to room temperature, the layers were separated and the aqueous layer was diluted and extracted with $3 \times 20$ mL of dichloromethane. The combined organic layers were washed with brine, dried over $Na_2SO_4$, filtered, and concentrated under reduced pressure to produce 120 mg of yellow oil. The crude was purified by flash chromatography on silica gel (petroleum ether/$CH_2Cl_2$ from 100/0 to 90/10) to provide 46 mg (36%) of white solid. $^1$H NMR (500 MHz, CDCl$_3$): 7.95 (s, 6 H), 7.91–7.73 (m, 26 H), 7.54 (s, 2 H), 7.46 (d, 2 H), 741–7.35 (m, 12 H), 2.08–2.05 (m, 24 H), 1.16–1.10 (m, 72 H), 0.81–0.74 (m, 60 H).

### 4.1.3. Synthesis of the Platinum Complex "C"

In a Schlenk tube, under argon atmosphere, ligand TFb (400 mg, 0.36 mmol) and PtCl$_2$(PBu$_3$)$_2$ (120 mg, 0.18 mmol) were inserted. Triethylamine (8 mL) was added under stirring, and the solution turned yellow. Then, CuI (5 mg, 0.03 mmol) was added. The disappearance of the initial ligand was checked using TLC (eluent: Cyclohexane and one drop of diethyl ether). The solvent was evaporated under a vacuum. The obtained solid was washed with distilled water, then extracted with diethyl ether. The ether phase was washed three times with distilled water and dried over MgSO$_4$. The ether was evaporated under vacuum. The powder was dissolved in a mixture of petroleum ether and isopropanol and let to evaporate at room temperature for crystallization as pale yellow crystals (418 mg, yield 82%). FTIR (cm$^{-1}$): $\nu(-C \equiv C-)$ 2100. $^1$H NMR (300 MHz, Aceton-$d_6$): 8.03 (s, 6 H), 7.96–7.74 (m, 24 H), 7.49–7.29 (m, 16 H), 2.25 (m, 12 H), 2.17 (m, 24 H), 1.77 (m, 12 H), 1.54 (m, 12 H), 1.11 (m, 72 H), 0.99 (t, 18 H) 0.75 (m, 60 H). $^{31}$P NMR (81.02 MHz, Aceton-D6): 3.99 ppm (s).

Synthesis of ((7-(3,5-Bis(9,9-dihexyl-9H-fluoren-2-yl)phenyl)-9,9-dihexyl-9H-fluoren-2-yl)ethynyl) trimethylsilane

To a degassed solution of 2,2′-(5-(9,9-dihexyl-7-iodo-9H-fluoren-2-yl)-1,3-phenylene)bis(9,9-dihexyl-9H-fluorene) (740 mg, 0.61 mmol) in THF (8 mL) and triéthylamine (8 mL), trimethylsilylacetylene (177 mg, 1.80 mmol), then bis(triphenylphosphine)palladium(II) dichloride (14 mg, 0.02 mmol) and iodide copper (8 mg, 0.04 mmol) were added. The mixture was stirred at 60 °C for 12 h, then concentrated under reduced pressure to produce 1 g of brown dark oil. The crude was purified by flash chromatography on silica gel (9/1 petroleum ether/$CH_2Cl_2$) to provide 614 mg (85%) of white solid. $^1$H NMR (300 MHz, CDCl$_3$): 7.90 (s, 1 H), 7.89 (s, 2H), 7.84–7.66 (m, 12 H), 7.50–7.47 (m, 2 H), 7.38–7.32 (m, 6 H), 2.06–2.00 (m, 12 H), 1.25–1.07 (m, 36 H), 0.78–0.69 (m, 30 H), 0.30 (s, 9 H).

Synthesis of 2,2′-(5-(7-Ethynyl-9,9-dihexyl-9H-fluoren-2-yl)-1,3-phenylene)bis(9,9-dihexyl-9H-fluorene)

Potassium carbonate (180 mg, 1.30 mmol) was added to a solution of ((7-(3,5-bis(9,9-dihexyl-9H-fluoren-2-yl)phenyl)-9,9-dihexyl-9H-fluoren-2-yl)ethynyl)trimethylsilane (580 mg, 0.50 mmol) in 5 mL of THF and 5 mL of methanol. The mixture was stirred for 2.5 h at room temperature and concentrated under reduced pressure. $CH_2Cl_2$ was added and the organic layer was washed twice with water, dried over $Na_2SO_4$, filtered, and concentrated over reduced pressure. The crude was

purified by flash chromatography on silica gel (petroleum ether/$CH_2Cl_2$ from 95/5 to 90/10) to provide 470 mg (86%) of white solid. [1]H NMR (300 MHz, $CDCl_3$): 7.91 (s, 1 H), 7.90 (s, 2 H), 7.84–7.81 (m, 3 H), 7.77–7.69 (m, 9 H), 7.53–7.50 (m, 2 H), 7.39–7.32 (m, 6 H), 3.16 (s, 1 H), 2.06–2.01 (m, 12 H), 1.31–1.07 (m, 36 H), 0.78–0.69 (m, 30 H).

### 4.2. Optical Absorbance and Luminescence

The linear absorbance, luminescence spectra, and life-times were measured following standard procedures as described by Glimsdal et al. and Lind et al. [29,41].

The NIR 2PA cross-sections for molecules "A" and "C" were measured from the fluorescence and phosphorescence, respectively, in Ar-purged solutions [28,42]. Due to its low luminescence, the 2PA cross-section of molecule "B" was not measured.

### 4.3. Wavelength-Scanned OPL Measurements

The wavelength-scanned OPL measurements were made in an $f$/5 setup [36,43]. The laser pulse energy was controlled by two filter wheels in OD0.1 increments, then a cleaned-up, near top-hat, 2 cm diameter laser beam was focused by a $f$ = 100 mm lens into the middle of the sample.

The beam cleanup was performed by a 3.33 Galilean beam expander, ≥25 μm 2x $f$ = 30 cm pinhole-filter, a second 2.66x Galilean beam expander, and finally, a 2 cm diameter iris in front of the $f$/5 lens.

The energy of each individual pulse is known because a pellicle beam-splitter in front of the 2.66x Galilean beam expander was used to sample the beam by an Ophir PE9-C (reference detector). The pulse energy on target was linear fit calibrated by placing an Ophir PE10-C at the sample position.

The light transmitted through the sample was collimated by an $f$ = 4 mm lens, passed through an 8.1 mm diameter aperture, focused by a $f$ = 1 m lens onto a 1.5 mm diameter iris, and measured by a Ophir PE10-C at a further 0.5 m distance (signal detector). It was linearly calibrated using the reference detector calibration, while measuring without a sample. Both the reference and signal detector were calibrated for each wavelength (Table S1).

The laser source was an EKSPLA NT342C OPO with 3–5 ns pulse length. The laser wavelength was scanned by ~25 nm increments between 450 nm and 700 nm. Some deviations from this pattern were made due to the varying reflectivity of the interference-based pellicle beam-splitter. The system was optically aligned at 450 nm.

For each wavelength, the horizontal 10–90% diameter and depth position of the beam were measured by the knife-edge method (Table S1). The beam diameter varied between 7.7 μm and 10.40 μm. A CMOS camera near the focus was used to ascertain the flatness of the beam. At 650 nm and longer, some weak skewness of the beam cross-section was noticed. The symmetry of the beam in the focus was investigated by microscopic burn marks on photographic film at 450 nm and 700 nm. The marks were symmetrically circular. At the higher fluences required to mark the photographic film at 700 nm, some spots showed weaker ghost spots at 2/3 of the spot diameter distance from the center. This is attributed to intermittent plasma air-popping in the focus of the beam cleanup Fourier filter at high fluences.

After the calibrations, the OPL measurements were performed. Each sample was a 1 mm quartz cuvette with 25 mM of either molecule "A", "B", or "C" in $CH_2CL_2$. At each wavelength, the $f$/5 lens was transposed depth-wise so that the beam cleanly passed through the 1.5 mm iris, in effect cancelling most chromatic aberrations. For the measurements with sample cuvettes, the $f$ = 4 mm lens had to be slightly transposed sideways due to the cuvette holder inducing a minute amount of sideways shift of the beam. At 450 nm, 560 nm, and 660 nm, the sample centering (depth-wise) was checked by performing a z-scan. For each wavelength, the pulse energy was increased in OD0.1 increment. The pulse rate was limited to 1 Hz to avoid boiling of the solvent, and the measurements were performed with manually triggered 3 pulse trains. To avoid fouling of the cuvette surface, the cuvette was translated 100 μm horizontally between each wavelength.

At high fluences, some indications of nonlinear refraction/scattering were found in the form of blooming of the beam at the signal detector. The *f*/5 measurement setup is not designed to distinguish between nonlinear refraction/scattering and reverse saturable absorption, so other techniques such as ultrafast spectroscopy had to be used instead [1,36]. At 475 nm, 575 nm, and 660 nm, measurements on a cuvette containing solely $CH_2Cl_2$ were made to search for any OPL from the solvent. For these baseline measurements, an OD1 filter was placed in front of the signal detector to increase the dynamic range.

### 4.4. Excited State Absorption

The microsecond ESA measurements were carried out as described by Glimsdal et al. [30]. Ar-purged THF was used as a solvent. A broadband white μs light source was used in conjunction with a 355 nm ns pulsed laser. By using a second flashlamp with tunable wavelength as a pump, it was possible to measure the lifetime of the excited triplet state for molecules "A" and "B" in Ar-purged THF solvent [44].

### 4.5. Ultra-Fast Transient Absorption Measurements

The experimental setup for recording broadband transient absorption spectra was based on a Ti:Sapphire amplified laser system (Spitfire XP Pro, Spectra Physics, Santa Clara, CA, USA) operating at a 1 kHz repetition rate, generating ~80 fs pulses with a central wavelength of 796 nm. The pump beam was tuned by an optical parametric amplifier (TOPAS C, Light Conversion, Vilnius, Lithuania) to excite the sample at 337 nm with typical fluency not exceeding $10^{14}$ photons per pulse per $cm^2$. A white-light supercontinuum was used as the broadband probe beam, generated by focusing the NIR signal from TOPAS C into a 5 mm sapphire plate. The desired timing between excitation and probe pulses was achieved by a computer-controlled delay line (10 ns, Aerotech; Pittsburgh, PA, USA). The pump and the probe beams were overlapped on the sample with their relative polarization set to the magic angle (54.7°) by a Berek polarization compensator placed in the pump beam path. The sample was placed in a quartz 1 mm path length cuvette with an automated sample mover to avoid possible sample photodamage, which was checked by measuring absorption spectra of the sample before and after each experiment. The probe and reference beams were collimated on the entrance aperture of a prism-based, double-beam spectrograph, and detected by a double diode-array detection system (Pascher Instruments, Lund, Sweden).

## 5. Conclusions

The three molecules investigated in this article all show good OPL performance through the visible wavelength range. Combined with their linear transparency, it makes them candidates for incorporation into glass materials, which is necessary for practical applications.

Their complex decay structure and OPL performance both with and without a central ISC promoter invite further study to gain a better understanding of design patterns for OPL molecules. The OPL performance of the molecule without an ISC promoter was better than for the molecule with a dibromophenyl ISC promoter due to the strong singlet ESA of the former being negated by a too rapid depopulation of the S1 state by ISC in the latter.

Combining singlet ESA and triplet ESA might be a fruitful design pattern for improving the performance of nanosecond OPL molecules. This, since the transmission during the beginning of the pulse, has the most impact for nanosecond pulses [36].

**Supplementary Materials:** The following are available online at http://www.mdpi.com/2304-6740/7/10/126/s1, Table S1: Calibration factors and beam width, Figure S1: OPL curves, 450 and 475 nm, Figure S2: OPL curves, 490 and 525 nm, Figure S3: OPL curves 560 and 575 nm, Figure S4: OPL curves, 590 and 625 nm, Figure S5: OPL curves, 650 and 660 nm, Figure S6: OPL curves, 700 nm, Figure S7: Transient absorption spectrum and fit for molecule "A" at selected wavelengths, Figure S8: Transient absorption spectrum and fit for molecule "B" at selected wavelengths, Figure S9: Transient absorption spectrum and fit for molecule "C" at selected wavelengths.

**Author Contributions:** C.L., S.P., C.A., T.P., and M.L. contributed to the general choice of materials and conceptualization; H.L. to development of methodology; D.P., J.-C.M., C.M., and A.L. to the synthesis of new materials; D.K.H. and M.L. to spectroscopic characterization; I.M. and P.C. to conceptualization of ultrafast measurements; and finally, H.L., M.L., P.C. and I.M. to measurements, data analysis, and drafting of the manuscript.

**Funding:** This work was supported by the Swedish Armed Forces.

**Acknowledgments:** Pontus Köhler is acknowledged for helping with the alignment of the OPL measurement setup. Jens Uhlig is acknowledged for giving a very helpful primer on curve-fitting for transient absorption data.

**Conflicts of Interest:** The authors declare no conflict of interest.

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
