# Peer review of "An Optical Power Limiting and Ultrafast Photophysics Investigation of a Series of Multi-Branched Heavy Atom Substituted Fluorene Molecules"

_inorganics, doi:10.3390/inorganics7100126_

Round 1
Reviewer 1 Report
The authors show the OPL performances of three molecules. They performed some photophysical analysis to understand the different performances of the molecules. The subject is interesting and the use of ultrafast spectroscopy can really help to understand the molecules performances. Unfortunatelly the analysis on the ultrafast transient absorption measurements are really confused and not well explained. They show the spectral basis vectors in the main text for the three molecules without showing the transient spectra at different probe delays and the most important temporal decays (they are supposed to be shown in the Supplementary Material but unfortunatelly I was not able to access to it). For my point of view these figures should be in the main text. In the discussion section they should comment in details the origin of the positive and negative bands present in the ultrafast absorption spectra and not just indicate them as ESA. Moreover at page 8 line 207 it is written " 545ns", while in the figure it is reported 545ps , which one is the right one?
In conclusion the paper can be publshed just after major revisions.
Author Response
The authors show the OPL performances of three molecules. They performed some photophysical analysis to understand the different performances of the molecules. The subject is interesting and the use of ultrafast spectroscopy can really help to understand the molecules performances. Unfortunatelly the analysis on the ultrafast transient absorption measurements are really confused and not well explained. They show the spectral basis vectors in the main text for the three molecules without showing the transient spectra at different probe delays and the most important temporal decays (they are supposed to be shown in the Supplementary Material but unfortunatelly I was not able to access to it). For my point of view these figures should be in the main text.Response: The supplementary information file has been fixed so that important information of transients in Figs S7-S9 is shown.
For my point of view these figures should be in the main text. In the discussion section they should comment in details the origin of the positive and negative bands present in the ultrafast absorption spectra and not just indicate them as ESA.Response: Line 214-219 in the updated version have more arguments and discussion about the positive-negative bands in the transient absorption.
Moreover at page 8 line 207 it is written " 545ns", while in the figure it is reported 545ps , which one is the right one?Response: This and similar typos have been fixed in the updated version.
Reviewer 2 Report
The authors investigated optical power limiting (OPL) performance in multi-branched fluorene molecules where the central moiety was either a pure organic, a dibromo-substituted unit or a platinum(II)-containing unit. First, Introduction must be implemented taking into account that all the examined systems contain a fluorene units. The introduction discussion lacks contextual insights and details justifying the insertion of such emissive units in the specific context. Moreover, in Pf. 2.1 Luminescence properties of the A, B and C compounds are discussed. A brief glimpse on luminescence properties of that class of compounds must be added both in Introduction and in Results sections. A link with OPL properties should be more clearly established. A useful comparison between purely organic materials (A and B) with metal containing (C) ones about emission performance should be added, referring to the difference between organic and metal-assembly fluorophore. More literature should be introduced regarding the effect of the metal on the PL performance, as for example “Photophysical Properties of Luminescent Zinc(II)‒Pyridinyloxadiazole Complexes and their Glassy Self-Assembly Networks” EUROPEAN JOURNAL OF INORGANIC CHEMISTRY. Vol. 2018. Pag.2709-2716. Other literature about the specific role of the platinum respect to other heavy metal should be added and discussed. An exhaustive discussion based on OPL measurements and on absorption in the excited state constitutes the main part and it is better focused and linked to the context. In Sect. 3. Discussion, lines 213-215: “We interpret this as the Br atoms facilitating inter-system 214 crossing while shortening the lifetime of the S1 state” should be added literature data to support. Lines 230-231: this statement seems risky unless experimentally justification supported by literature data. In the Conclusion differences between OPL performances of the three molecules must be detailed. Melting point of the new compounds must be added. The manuscript is well written in proper English. It actually provides results in the field of OPL materials, focusing interesting experimental/theoretical data. At this stage the paper can be considered for publication on Inorganics after the recommended revision.Author Response
The authors investigated optical power limiting (OPL) performance in multi-branched fluorene molecules where the central moiety was either a pure organic, a dibromo-substituted unit or a platinum(II)-containing unit.
First, Introduction must be implemented taking into account that all the examined systems contain a fluorene units. The introduction discussion lacks contextual insights and details justifying the insertion of such emissive units in the specific context. Moreover, in Pf. 2.1 Luminescence properties of the A, B and C compounds are discussed. A brief glimpse on luminescence properties of that class of compounds must be added both in Introduction and in Results sections. A link with OPL properties should be more clearly established. A useful comparison between purely organic materials (A and B) with metal containing (C) ones about emission performance should be added, referring to the difference between organic and metal-assembly fluorophore. More literature should be introduced regarding the effect of the metal on the PL performance, as for example “Photophysical Properties of Luminescent Zinc(II)‒Pyridinyloxadiazole Complexes and their Glassy Self-Assembly Networks” EUROPEAN JOURNAL OF INORGANIC CHEMISTRY. Vol. 2018. Pag.2709-2716. Other literature about the specific role of the platinum respect to other heavy metal should be added and discussed.
Response: We have added some justification for using a platinum complex and some reference to their luminescent properties in lines 71-73 of the new version. There is many references to Pt-complexes throughout the manuscript including our own work. Here is more reference and discussion to dibromine complexes and their photophysics (refs 10 and 11) around line 209 in new version.
The reference the reviewer suggests is interesting but we feel its main topic/content is not very relevant for our investigation. Our paper primarily deals with OPL application and ISC assisted by metal ions, not primarily the luminescent properties.
An exhaustive discussion based on OPL measurements and on absorption in the excited state constitutes the main part and it is better focused and linked to the context.
In Sect. 3. Discussion, lines 213-215: “We interpret this as the Br atoms facilitating inter-system 214 crossing while shortening the lifetime of the S1 state” should be added literature data to support. Lines 230-231: this statement seems risky unless experimentally justification supported by literature data. In the Conclusion differences between OPL performances of the three molecules must be detailed.
Response: We have added more discussion and reference to Br compounds (see above). There is also added more discussion with reference to transient absorption of the Br compound in lines 214-219 in the updated version and in the conclusions.
Melting point of the new compounds must be added.
Response: We understand from the suggested reference of European Journal of Inorganic Chemistry that the reviewer is interested in such data, but we did not record it as we used the compounds solely with organic solvents.
The manuscript is well written in proper English. It actually provides results in the field of OPL materials, focusing interesting experimental/theoretical data. At this stage the paper can be considered for publication on Inorganics after the recommended revision.
Reviewer 3 Report
Lunden et al in this manuscript describe three multi-branched fluorene analogs and investigate their OPL properties. Different ISC promoter moieties have been chosen and the authors found all three of them to have good OPL properties in the visible range. It is recommended that combining singlet ESA and triplet ESA could lead to a better prototype for OPL molecules. Author explain their findings in a very detailed fashion. Their experiential results conveyed the message in detail. All the data looks promising and I would recommend its publication in Inorganics after the minor comments below are addressed.
Line 80: Fix the typo in “stae”. Line 186: Make figure 8 consistent with Figure 6 and 7. Line 192: The authors have an incomplete sentence “ The lowered cut-off level towards 700 nm for molecule 'C' is probably explained by.” Lines 196/ Line 203: Try to combine these two paragraphs and avoid redundancy. I found this was the only paragraph in the conclusion section that was not to the point. Line 205/ 213: Unnecessary punctuations need to be fixed (pronounced "PA efficie../ to molecule 'A's 545 ps decay).Author Response
Reply to reviewer 3:
Lunden et al in this manuscript describe three multi-branched fluorene analogs and investigate their OPL properties. Different ISC promoter moieties have been chosen and the authors found all three of them to have good OPL properties in the visible range. It is recommended that combining singlet ESA and triplet ESA could lead to a better prototype for OPL molecules. Author explain their findings in a very detailed fashion. Their experiential results conveyed the message in detail. All the data looks promising and I would recommend its publication in Inorganics after the minor comments below are addressed.
Line 80: Fix the typo in “stae”.
Response: We have fixed this and other typos.
Line 186: Make figure 8 consistent with Figure 6 and 7.
Response: We have replotted Figure 8 to conform with Figs 6 and 7.
Line 192: The authors have an incomplete sentence “ The lowered cut-off level towards 700 nm for molecule 'C' is probably explained by.”
Response: The missing ‘2PA’ has been reinserted.
Lines 196/ Line 203: Try to combine these two paragraphs and avoid redundancy. I found this was the only paragraph in the conclusion section that was not to the point.
Line 205/ 213: Unnecessary punctuations need to be fixed (pronounced "PA efficie../ to molecule 'A's 545 ps decay).
Response: We have fixed these typos and errors.
Round 2
Reviewer 1 Report
The authors have answered to all my comments. The paper can now be published.